# Review: Influence of 25(OH)D Blood Concentration and Supplementation during Pregnancy on Preeclampsia Development and Neonatal Outcomes

**DOI:** 10.3390/ijms232112935

**Published:** 2022-10-26

**Authors:** Nataliia Karpova, Olga Dmitrenko, Ekaterina Arshinova, Malik Nurbekov

**Affiliations:** Federal State Budgetary Institution “Research Institute of Pathology and Pathophysiology”, Moscow 125315, Russia

**Keywords:** preeclampsia, neonatal outcome, 25-hydroxyvitamin D, 25(OH)D, 1,25(OH)2D, diabetes, pregnancy complication, vitamin D deficiency, vitamin D supplementation

## Abstract

Briefly, 25-hydroxyvitamin D (25(OH)D) plays an essential role in embryogenesis and the course of intra- and postnatal periods and is crucially involved in the functioning of the mother–placenta–fetus system. The low quantity of 25(OH)D during pregnancy can lead to an elevated risk for preeclampsia occurrence. Despite the numerous studies on the association of 25(OH)D deficiency and preeclampsia development, the current research on this theme is contradictory. In this review, we summarize and analyze study data on the effects of 25(OH)D deficiency and supplementation on pregnancy, labor, and fetal and neonatal outcomes.

## 1. Preeclampsia and 25(OH)D Insufficiency/Deficiency

Preeclampsia is a multisystem pathological condition that occurs after the 20th week of pregnancy and is characterized by arterial hypertension combined with proteinuria (≥0.3 g/L in daily urine), edema, and manifestations of multiple organ dysfunction [1,2,3]. Preeclampsia occurs in 3–8% of pregnant women and is one of the top five causes of maternal morbidity and mortality [4,5,6,7,8].

The etiology and pathogenesis of preeclampsia are not fully understood. Placental hypoxia and/or ischemia, high levels of oxidative stress, endothelial dysfunction, and trophoblast immaturity are the hallmarks of preeclampsia. Moreover, vascular reactivity is caused by an imbalance and malfunction in the chemicals that cause vasodilation and vasoconstriction. The ensuing vasoconstriction reduces mother organ perfusion and placental perfusion, which might harm end organs. There are potential epigenetic, biochemical, and biophysical biomarkers for predicting the development of this pathology. Numerous studies have confirmed the contribution of 25-hydroxyvitamin D (25-Hydroxycalciferol, 25(OH)D) concentrations to the development of preeclampsia and associated complications, such as endothelial and vascular dysfunction [2,3,9,10,11,12,13,14,15,16,17,18,19,20,21,22,23,24,25,26].

Studies show a correlation between 25(OH)D concentration levels and preeclampsia severity until the 22nd week of gestation [11,27,28]. However, it is not clear whether the level of 25(OH)D is depleted due to the preeclampsia development or whether the low level of the 25(OH)D causes the preeclampsia to progress. It is also essential to determine whether the 25(OH)D level has a role in early-onset preeclampsia development or can act as a marker of it. In this review, we will analyze and discuss all the significant studies connected with the role of 25(OH)D deficiency in preeclampsia development.

Early scientific evidence on the relationship between preeclampsia and 25(OH)D was contradictory—a decrease in preeclampsia progression was associated with 25(OH)D administration, while the supplementation addition (such as calcium or halibut oil) was not linked with the treatment and prevention of the disease. Recent studies have shown that low 25(OH)D concentrations are an independent risk factor for early- and late-onset disease [29,30,31,32,33,34]. For example, Ullah et al. found up to a five-fold increase in the risk of developing preeclampsia and eclampsia in women with 25(OH)D deficiency [35].

For preeclampsia research, most of the measurements of 25(OH)D concentrations are collected from the blood of pregnant women. Halhali et al. (2007) refreshed and simplified this approach by showing that total urinary calcium excretion positively correlated with blood 1,25-dihydroxyvitamin D (1,25(OH)2D) concentration in both normotensive and preeclamptic women [36]. On the other hand, Hyppönen et al. investigated the relationship between 25(OH)D intake at an early age and the subsequent development of preeclampsia during pregnancy and found that the risk of preeclampsia was halved in women who received regular 25(OH)D supplements during the first year of life [37]. Bener et al. also noted a significant relationship between maternal 25(OH)D deficiency during pregnancy with an increased risk of not only preeclampsia but gestational diabetes mellitus (GDM) and anemia [38]. Scholl et al. found that some women with 25(OH)D deficiency developed secondary hyperparathyroidism, which was associated with an increased risk of preeclampsia [39].

Recent studies investigated the role of 25(OH)D concentration in gestational hypertension or preeclampsia [40,41]. In the context of preeclampsia, it is important to consider concentrations of the active form of 25(OH)D, 1,25(OH)2D. The enzyme 25-hydroxyvitamin D-1 alpha-hydroxylase (1 alpha-OHase) converts 25(OH)D to 1,25(OH)2D. In the preeclamptic placenta, this enzyme is significantly reduced, which can lead to insufficient concentrations of the active form of 25(OH)D [42]. According to available data, low concentrations of 25(OH)D do not necessarily lead to low concentrations of 1,25(OH)2D [34]. However, in a study by Abbasalizadeh et al. a mediated effect of 25(OH)D deficiency on the risk of developing preeclampsia through hypocalcemia was found, which increases the risk by 8.5 times [41].

These data support the existence of a threshold level of 25(OH)D concentration, below which, the risk of preeclampsia may increase, both directly through the regulation of gene expression and indirectly through changes in calcium metabolism.

## 2. Concentrations of 25(OH)D in the First Trimester of Pregnancy (<14 Weeks) Complicated by Preeclampsia

For the early prediction and diagnosis of preeclampsia, a number of researchers have sampled and analyzed whether 25(OH)D concentrations can affect the development of pathology in the first trimester of pregnancy. Results have shown that low blood 25(OH)D concentrations in the first trimester of pregnancy are not associated with adverse pregnancy outcomes and do not predict any complications [39,43,44,45,46,47]. Although 25(OH)D concentrations were not associated with preeclampsia, Benachi et al. showed that women with 25(OH)D concentrations within the normal range in the first and third trimesters of pregnancy had a significantly lower risk of developing preeclampsia. It emphasizes the hypothesis that there is a threshold effect of 25(OH)D concentration on the risk of developing preeclampsia: the risk of preeclampsia increases with 25(OH)D concentration < 30 ng/mL (1 ng/mL = 0.400641 nmol/L) [47]. This hypothesis is also supported by the results of Achkar et al. 2015, where 25(OH)D deficiency early in pregnancy (before 14 weeks), defined as 25(OH)D3 < 30 nmol/L, was an independent risk factor for the development of preeclampsia [48].

Overall, the data demonstrate that solely 25(OH)D concentrations cannot be used to forecast preeclampsia. However, women with a 25(OH)D3 deficiency < 30 ng/mL are at increased risk and need to be prescribed supplements to normalize blood 25(OH)D concentrations.

## 3. Concentrations of 25(OH)D in the Second Trimester of Pregnancy (14–26 Weeks) Complicated by Preeclampsia

Although there is no direct relationship between 25(OH)D concentrations and preeclampsia during the second trimester, decreased maternal blood 25(OH)D concentration was found to be associated with an increased risk of preeclampsia [49,50,51,52,53]. However, some researchers have found no correlation [54,55].

## 4. Concentrations of 25(OH)Din the Third Trimester of Pregnancy (>26 Weeks) Complicated by Preeclampsia

During the third trimester, total free (total minus DBP- and albumin-bound) and bioavailable (total minus DBP-bound) maternal 25-hydroxyvitamin D3 (25(OH)D3) correlated with placental 25(OH)D3 in the third trimester, but this trend did not persist in preeclampsia [56,57]. Tamblyn, J A et al. showed that women with preeclampsia had significantly lower concentrations of 1,25(OH)2D3 and albumin than women with a normal pregnancy. However, concentrations of 25(OH)D3 were not significantly different. In contrast, women with preeclampsia had the highest blood concentrations of 3-epi-25(OH)D3 and 24,25(OH)2D3. The levels of 25(OH)D3 and 3-epi-25(OH)D3 in the placenta of women with preeclampsia were lower than in physiologic pregnancy, while 24,25(OH)2D3 concentrations were the highest in the preeclamptic placenta. In conclusion, preeclampsia is associated with decreased activation, increased catabolism, and impaired placental 25(OH)D3 uptake [57].

While comparing the results of measuring the concentration of 25(OH)D throughout pregnancy by different researchers, it was discovered that the concentration of 25(OH)D < 30 ng/mL in women with preeclampsia (Table 1). These results indicate the need to maintain an optimal level of 25(OH)D throughout the entire gestation period.

## 5. Impact of 25(OH)D Concentrations on the Development of Arterial Hypertension during Pregnancy Complicated by Preeclampsia

Concentrations of 25(OH)D play an essential role in the regulation of blood pressure (BP), affecting vascular endothelial function, oxidative stress, and the placenta [34,58]. In addition, 25(OH)D regulates key target genes associated with placental invasion, normal implantation, and angiogenesis [34,42]. Thus, the level of 25(OH)D may be one of the causes of arterial hypertension in preeclampsia. According to several studies, low concentrations of 25(OH)D and 1,25(OH)2D correlate with high systolic and diastolic BP during pregnancy [59,60,61,62,63,64], and only a few papers have demonstrated the opposite results [65]. Because 25(OH)D is an approved supplement during pregnancy, Nassar, Seham Zakaria, and Noha Mohamed Badae tested the hypothesis of lowering systolic BP by 25(OH)D supplementation and found that its administration reduced systolic BP and proteinuria [66]. Thus, there is strong evidence for the need to measure 25(OH)D concentrations among pregnant women with any form of hypertension to prescribe optimal doses of 25(OH)D, thereby possibly providing a reduction in elevated systolic and diastolic blood pressure, reducing the risk of additional complications for the mother and fetus.

## 6. Impact of Proteinuria on 25(OH)D Concentrations during Pregnancy Complicated by Preeclampsia

One of the key symptoms of preeclampsia is proteinuria–vitamin D-binding protein (DBP) loss in the urine. For this reason, women with preeclampsia form a vulnerable 25(OH)D deficiency group, the reabsorption of which occurs in the proximal renal tubules after endocytosis together with the 25(OH)D-binding protein, because proteinuria contributes to damage and tubular dysfunction. Albejante et al. suggested that the proteinuria seen in preeclampsia may contribute to the loss of reabsorption of 25(OH)D along with other proteins in the urine. As a result, the authors were able to prove an association between preeclampsia and 25(OH)D deficiency, because proteinuria causes the loss of proteins responsible for the reabsorption of 25(OH)D by renal tubules. In combination with other factors, proteinuria can accelerate 25(OH)D deficiency in preeclampsia [67]. Therefore, both the symptoms of preeclampsia, arterial hypertension and proteinuria, are associated with 25(OH)D deficiency. In addition, the optimal 25(OH)D (>30 ng/mL) concentrations before pregnancy can have a small impact on immunity and embryogenesis, while an initial deficiency and an additional loss of 25(OH)D can cause serious consequences, especially for fetal development.

## 7. Relationship between 25(OH)D Concentrations and the Risk of Preeclampsia in Diabetes Mellitus

The 25(OH)D metabolism changes rapidly during pregnancy. When it results in deficiency, it starts associating with a number of adverse complications, including gestational diabetes and preeclampsia [68,69,70,71]. Pregestational diabetes type 1 or type 2 is a risk factor for preeclampsia. Preeclampsia is known to be diagnosed in 15–20% of pregnancies in women with type 1 diabetes and in 10–14% of pregnancies in women with type 2 diabetes, and the incidence of preeclampsia in gestational diabetes mellitus (7.3%) is higher than in the general population (4.5%) [71].

The results of studies of the relationship between the 25(OH)D status in a pregnant woman with type 1 diabetes and preeclampsia show that lower 25(OH)D concentrations were observed in women with preeclampsia due to type 1 diabetes compared with women with type 1 diabetes alone [72,73,74,75]. Several studies support an association between low blood 25(OH)D levels and an increased risk of GDM and preeclampsia [69,70]. In contrast, a recent case–control study found no evidence of an association between the 25(OH)D levels in either the former or in the second half of pregnancy and the further development of GDM [76].

Despite the high prevalence of type 2 diabetes in pregnant women, the association with changes in 25(OH)D levels in pregnant women with type 2 diabetes and preeclampsia has not yet been studied [77]. Given the increasing prevalence of pregestational diabetes and GDM in pregnant women, future studies should continue to investigate the impact of 25(OH)D deficiency on the risk of developing preeclampsia in these pathologies.

## 8. Maternal 25(OH)D Concentrations after Delivery

While the conversion of 25(OH)D to 25(OH)D remains unchanged during pregnancy, the conversion of 25(OH)D to 1,25(OH)2 D during pregnancy is unique. Recent research data indicate that starting early in pregnancy, serum 1,25(OH)2 D concentrations increase and reach a three-fold increase in the third trimester. Moreover, 1,25(OH)2 D levels may be influenced by the placenta, which contains the enzyme CYP27B1 (25-hydroxyvitamin D3 1-alpha-hydroxylase) and produces 1,25(OH)2 D [78,79].

## 9. The Influence of Maternal 25(OH)D Concentrations on the Health of Newborns and Young Children

Concentrations of 25(OH)D and the optimal amount of its metabolites contribute to the development of the fetus and minimize the risk of adverse pregnancy outcomes. Depleted levels of 25(OH)D in the body of a pregnant woman contribute to an increase in the incidence of health complications, not only in the mother but also in newborns [11,80,81], including from pre-eclamptic mothers [50,80,81,82,83,84,85,86,87,88,89,90,91,92,93]. Study shows more than half of the children born prematurely have 25(OH)D deficiency (less than 20 ng/mL) [94]. The protective effect of 25(OH)D has been demonstrated in a study by Vasilyeva et al., 2017, showing a reduction in adverse neonatal outcomes such as fetal growth retardation, hypoxia, fetal cerebral injury, and 25(OH)D deficiency with 25(OH)D supplementation at a concentration of 4000 IU/day [95].

The insufficient amount of 25(OH)D is associated with many adverse neonatal outcomes: 25(OH)D deficiency in newborns [11,80,81], births premature and small for gestational age children [50,88,89,90,91,92,93,94,95,96], low birth weight [93], the limitation of head growth, the length and weight of the fetus in the third trimester [93], Apgar score at 1 and 5 min < 7 points [97], respiratory syncytial infection in children at 1 year of age [98], the deterioration of respiratory status [99], and BPD in preterm infants [100], asthma [101] and childhood wheezing [102] (Table 2) (Figure 1).

Nevertheless, not all researchers have found an association between 25(OH)D deficiency and neonatal outcomes [103,104,105]. The following researchers, Velkavrh M, Wierzejska R, and Levkovitz O did not confirm the relationship between maternal 25(OH)D concentration and the weight, body length, and head circumference of the newborn [105,106,107]. Cooper et al. (2016) showed that the whole-body bone mineral content in infants born to mothers administered cholecalciferol 1000 IU/day was not significantly different from that in infants born to mothers administered with a placebo [108].

The deficiency of 25(OH)D during pregnancy impairs fetal skeletal formation and leads to reduced fetal bone mass, but also has a certain effect on the child’s susceptibility to diseases immediately after birth since 25(OH)D affects the synthesis of surfactant, and one of its properties is the ability to stimulate the formation of peptides (β2-defensins and cathelicidins) with pronounced bactericidal activity [109,110,111,112,113,114,115,116,117]. In addition, 25(OH)D can regulate both adaptive and innate immune responses at the fetal–maternal interface [114,115]. Briefly, 25(OH)D is a neurosteroid that is essential for the division, growth, and differentiation of neurons and has a neuroprotective and neurotrophic effect. For these reasons, an insufficient intake of 25(OH)D in the fetus causes a decrease in the levels of neurotrophic factors in the brains of newborns (NGF, p75NTR, GDNF]) and may lead to a delay in the formation of fetal brain structures [117,118].

The deficiency of 25(OH)D in pregnant women adversely affects the postnatal development of children, especially when the level of 25(OH)D in the mother’s blood serum is less than 50 nmol/L (<20 ng/mL) [50,83,115]. Furthermore, 25(OH)D deficiency in the early postnatal period may result in delayed speech development, fainting, epilepsy, and a number of demyelinating diseases [119,120,121]. In addition, it has been shown that preschool children with attention deficit hyperactivity disorder at birth had low levels of 25(OH)D3 in the cord blood compared to newborns without this pathology [122,123]. The deficiency of 25(OH)D at birth can cause bronchial asthma in children in their first 10 years of life [100,101]. A study by Thiele et al. (2017) showed a linear dependence of the dynamics of the physical development of children at the age of 18 months on the level of 25(OH)D in the umbilical cord blood [82].

The level of 25(OH)D3 in the cord blood of a child is 50–80% of its level in the mother’s blood serum, regardless of the gestational age [112]. Since 25(OH)D3 deficiency is widespread among pregnant women, this explains the large number of newborns with 25(OH)D3 deficiency by the appointment of 25(OH)D3 supplements during pregnancy. At the same time, it was noted that the most important factor that allows the concentration of 25(OH)D3 in the blood serum of pregnant women and, as a result, newborns, is the provision of 25(OH)D3 supplements during pregnancy [124,125,126].

## 10. Safety of Vitamin D3 Supplements Prescription during Pregnancy

The period of pregnancy is a high-risk state for both mother and fetus. It is important to conduct qualitative and quantitative research to prescribe any supplement, especially for hormone-like vitamin D3. As some preclinical animal studies show, there is potential dose-dependent toxicity of vitamin D3 to the fetus—growth failure, skeletal malformations, and cardiovascular anomalies. However, Roth (2011) found no data on the teratogenic effects of vitamin D3 on women during pregnancy and their newborns [127]. Magielda-Stola et al. stated that vitamin D3 supplementation is a safe treatment option with no risk of side effects or toxicity. Vitamin D3 supplementation should be given at the planning stage of pregnancy in order to receive the most beneficial effect, with serum 25(OH)D levels in excess of 30 ng/mL [128].

## 11. Effect of Vitamin D3 Supplementation on Pregnancy Complicated by Preeclampsia

Medical community suggests that vitamin D3 supplementation can help prevent preeclampsia and increase 25(OH)D levels to optimal levels, including in newborns [124,125,126]. Vitamin D3 supplementation was used as early as 1942. Taking 1200 IU/day of vitamin D3 with calcium (375 mg/day) for 8–10 weeks by a pregnant woman reduced blood pressure in women susceptible to preeclampsia, thereby reducing the incidence of pathology [129]. Numerous studies have demonstrated that vitamin D3 supplementation reduces the risk of preeclampsia [130,131,132,133,134,135,136].

The effect of vitamin D3 supplementation may vary depending on the initial maternal serum concentrations of the vitamin, the dosage administered, and the frequency of administration. A study by Zatollah et al. showed that the multimineral supplementation of vitamin D3 (800 mg calcium, 200 mg magnesium, 8 mg zinc, and 400 IU vitamin D3) for 9 weeks in pregnant women at risk of preeclampsia led to an increase in the circulating levels of calcium, magnesium, zinc, and 25(OH)D in maternal serum and a decrease in both systolic and diastolic blood pressure [137]. According to Maryam et al. after supplementation without the prior assessment of vitamin D3 levels, only 2% of women achieved sufficient 25(OH)D levels (>20 ng/mL) compared with 53% of women who participated in screening. Adverse pregnancy outcomes, including preeclampsia, gestational diabetes mellitus, and preterm birth, were reduced by 60%, 50%, and 40%, respectively, at the screening site. An additional injection of D3 at a dosage of 50,000 IU in addition to monthly maintenance therapy most contributed to the achievement of sufficient 25(OH)D concentration during labor [138]. These data suggest that optimal vitamin D concentrations, which can be achieved with supplementation, reduce the risk of disease. Moreover, in a study by Skowrońska-Jóźwiak et al. (2014), half of the subjects who reported taking vitamin D3 failed to reach the optimal serum 25(OH)D concentration [139].

Vitamin D3 supplementation can improve insulin sensitivity and glucose tolerance, which are the main pathophysiological disorders in diabetes (DM1, T2DM, GDM)—a risk factor for preeclampsia [140,141,142,143]. Samimi et al. noted that high-concentration vitamin D3 supplementation along with calcium (50,000 IU vitamin D3 every 2 weeks and 1000 mg calcium daily) for 12 weeks beneficially affected glycemic status, HDL-cholesterol, GDM incidence, and blood pressure among susceptible women with a risk of preeclampsia [58]. In another study, the administration of 50,000 IU 25(OH)D every 2 weeks from 20 to 32 weeks of gestation resulted in a significant increase in antioxidant system proteins and had a beneficial effect on lipid and insulin metabolism [144]. In a clinical trial where pregnant women at 12–16 weeks of gestation with a serum 25(OH)D concentration of less than 30 ng/mL received vitamin D3 supplements (50,000 IU) every 2 weeks, Mahdieh et al. 2015 noted a decrease in the incidence of GDM, but no difference was observed between neonatal outcomes [145].

Vitamin D3 supplementation during pregnancy also has a positive effect on neonatal outcomes. When taking low doses (≤2000 IU/day) of vitamin D3, there is a significant reduction in the risk of intrauterine or neonatal mortality, as well as the formation of a small fetus by gestational age [146]. Zatollah et al. (2015) found that a multimineral vitamin D3 supplement during pregnancy with the risk for preeclampsia resulted in an increase in neonatal body length [137]. Higher doses of vitamin D3 (1000–4000 IU/day) have been used by Faustino et al., who found that these doses may be convenient to achieve better outcomes for improving maternal, fetal, and subsequent offspring health [147]. A study by Nausheen et al. (2019) also found that vitamin D3 supplementation of 4000 IU per day was more effective in reducing vitamin D3 deficiency among pregnant women and in improving serum 25(OH)D levels in mothers and their newborns compared to 2000 IU per day and 400 IU per day [148]. A study by Vasilyeva et al. (2017) showed a comparative assessment of the condition of newborns from women at a high risk of preeclampsia with preconception preparation with vitamin D3 at a dose of 4000 IU/day in comparison with a no-supplementation-received group [95]. In women who received 4000 IU/day of vitamin D3, the level of 25(OH)D3 was 2.2 times higher (28.3 ± 1.5 ng/mL versus 12.4 ± 1.1 ng/mL). ml, respectively, *p* < 0.05) had newborns with a decreased frequency of clinical syndromes (fetal growth retardation, hypoxia, and cerebral lesions of the fetus) compared to the control group. However, not all researchers found a similar effect. Mirzakhani et al. showed that 25(OH)D levels of 30 ng/mL or higher at the beginning and at the end of pregnancy were associated with a lower risk of preeclampsia development, while vitamin D3 4400 IU started between 10 and 18 weeks of gestation did not reduce the incidence of preeclampsia [149].

It is important to consider that routine doses of vitamin D may be insufficient to achieve optimal levels (>30 ng/mL), and excessively high vitamin D concentrations may be hazardous to both maternal and fetal health. For this reason, it is suggested to measure vitamin D levels prior to prescribing a supplement to determine the appropriate dosage for each individual case.

In conclusion, the administration of vitamin D3—50,000 IU of vitamin D3 once every two weeks—appears to be protective against recurrent preeclampsia. Moreover, vitamin D3 therapy during pregnancy may help reduce the incidence of gestational hypertension [150].

## 12. Recommendations for the Prevention of Pregnancy Complications with Vitamin D3 Supplements

The currently existing national and international clinical recommendations for the intake of vitamin D3 during pregnancy differ significantly from each other. Various investigators have noted the need for vitamin D3 supplementation but have considered different dosages: 200 IU [151], 400 IU [152,153], 600 IU [154], 50,000 IU [144,145]. Grant (2010) concluded that the blood levels of 25(OH)D should be 80 nmol/L for optimal health. However, they found that 1000 IU of vitamin D3 per day increased serum 25(OH)D levels by about 10 ng/mL [155]. However, a meta-analysis of studies on the effect of 25(OH)D3 supplementation on the occurrence of pregnancy complications in the Cochrane database, considered the current “gold standard” of evidence-based medicine, did not answer the question of whether 25(OH)D3 supplementation should be used as a standard preparation for pregnancy planning [131]. The European Food Safety Authority (EFSA) states that the adequate intake of 25(OH)D3 for pregnant women remains the same as for non-pregnant women, at 600 IU per day [156]. These recommendations assume good nutrition and minimum effective exposure to the sun.

Recommendations for vitamin D supplementation to prevent pregnancy complications vary greatly from country to country. The International Endocrinological Society (clinical recommendations from 2011) recommends taking 1500–2000 IU per day, while the US Institute of Medicine (IOM) does not recommend >600 IU per day for pregnant women due to safety concerns. However, experts suggest that in the case of suboptimal 25(OH)D concentrations, it may be necessary to increase the daily dose to 1500–2000 IU with a maximum allowable dose of 4000 IU [157]. The Argentine Federation of Endocrinological Societies suggests 800–1200 IU per day. The new IOM and American Academy of Pediatrics proposal for children from birth to one year of age is 400 IU per day, and for those between 1 and 18 years of age, it is 600 IU per day [158]. According to Russian clinical guidelines, pregnant and lactating women need to take at least 800–1000 IU of vitamin D3 per day. When 25(OH)D deficiency is detected, the adequate correction of levels at a dose of 1500–4000 IU per day is necessary [159].

Although there is no consensus on the optimal dose of vitamin D3 during pregnancy, it is recommended to maintain optimal levels of 25(OH)D in the blood of pregnant women. The administration of the supplemental dosage should be calculated depending on the region of residence, lifestyle, and nutrition [38]. Recent scientific evidence suggests the relevance of using high doses (50,000 IU) of vitamin D3 every 2 weeks instead of weekly low doses of vitamin D3 (200–400 IU), but there is still a question about the duration of supplementation [138,144,145].

## 13. Conclusions

The results of numerous studies suggest that blood 25(OH)D concentrations alone cannot be used to predict preeclampsia. However, women with 25(OH)D deficiency <30 ng/mL are at increased risk and require supplementation to normalize blood 25(OH)D concentrations. The amount of severe neonatal outcomes and negative long-term consequences in children increases with maternal 25(OH)D concentrations < 20 ng/mL. Moreover, low concentrations of 25(OH)D and 1,25(OH)2D correlate with high blood pressure and proteinuria during pregnancy.

Insufficiency and deficiency of 25(OH)D in pregnant women can be compensated by supplementation containing Vitamin D3, with every 1000 IU of vitamin D3/day increasing serum 25(OH)D concentration by 10 ng/mL, and the use of large doses of vitamin D3 (50,000 IU) every 2 weeks reduces the risk of developing both preeclampsia and other pregnancy complications along with negative neonatal outcomes.

## Figures and Tables

**Figure 1 ijms-23-12935-f001:**
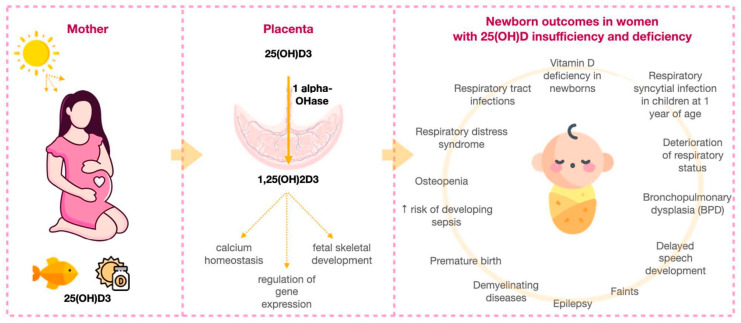
Neonatal outcomes in conditions of maternal 25(OH)D insufficiency and deficiency.

**Table 1 ijms-23-12935-t001:** Concentrations of 25(OH)D during normal and complicated by preeclampsia pregnancy.

Week of Gestation	25(OH)D Concentrations during Normal Pregnancy, mean ± SD (ng/mL)	25(OH)D Concentrations during Pregnancy Complicated by Preeclampsia, mean ± SD (ng/mL) ^2^	References
First trimester of gestation (<14)			
12	19.44 ± 8.2	20.88 ± 8.2	[44]
14	18.88 ± 7.08	20.92 ± 6.88	[48]
First–second trimester of gestation			
12–18	20.1 ± 9.3	22.3 ± 11.1	[47]
12–18	19.33 ± 4.75	pre-eclampsia 12.29 ± 2.79 severe pre-eclampsia 9.56 ± 2.68	[52]
Second trimester of gestation (15–26)			
15–20	39.2 (27.2–45.2) ^1^	30 (18.8–42.8) ^1^	[31]
15–21	28.6 ± 12.6	27.4 ± 14.4	[54]
16	18.84–23.96 (21.24) ^1^	18.16 (15.44–21.36) ^1^	[11]
24–26	22.8 ± 7.64	19.56 ± 6.72	[49]
Second–third trimester of gestation			
20–32	19.76 ± 9.04	16.92 ± 6.92	[57]
25–35	15.73 ± 5.85	10.09 ± 6.66	[27]
25–38	22.76 (17.56–28.32) ^1^	All preeclampsia 21.84 (18.2–27.84) ^1^	[43]
		Early onset preeclampsia 21.4 (16.96–24.68) ^1^	
25–39	14.9 ± 12.0	Severe pre-eclampsia 5.8 ± 4.5	[53]
		Non-severe pre-eclampsia 11.8 ± 7.3	
Third trimester of gestation (>26)			
26–31	32 (20–44) ^1^	18 (13–31) ^1^	[30]
26–37	13.41 ± 8.05	6.88 ± 9.46	[20]
30	10.09 ± 6.6	15.73 ± 5.85	[15]
30–40	23.7 ± 5.93	19.3 ± 4.31	[14]
30–40	19.5 ± 6.5	14.8 ± 5.4	[22]
30–40	31.4 ± 1.7	11.0 ± 7.1	[28]
30–36	22.57 ± 4.33	pre-eclampsia 18.68 ± 3.50 severe pre-eclampsia 9.48 ± 2.98	[52]
32–38	30.8 ± 11.0	27.7 ± 12.2	[47]
35–40	24.86 ± 1.02	23.96 ± 1.31	[35]
35–41	23.84 ± 6.93	15.27 ± 3.52	[19]

^1^ mean (min–max) 25(OH)D. ^2^ In the mentioned studies for Table 1, the 25(OH)D concentration measurements were performed with various methods like ELISA, ECLIA, chemiluminescent assay, and liquid chromatography coupled with mass spectrometry. Some of these methods like ELISA could give cross-reactions between 25(OH)D2 and 25(OH)D3 and could not distinguish these two forms (11). The difference between the kits and analysis equipment for chemiluminescent assays could also influence the precision of the obtained results. These data are presented in Appendix A.

**Table 2 ijms-23-12935-t002:** Concentrations of 25(OH)D threshold concentrations associated with adverse neonatal and postnatal outcomes.

Adverse Neonatal and Postnatal Outcomes	25(OH)D Threshold Concentrations in ng/mL ^2^	Reference
Neonatal Outcomes		
25(OH)D Deficiency in Newborns	<12	[80]
Low Body Mass at Birth	<30	[50]
Birth of Premature Babies	<20	[94]
	<30	[96]
Apgar score at 1 and 5 min < 7 points	<30	[97]
Deterioration of Respiratory Status	<12	[99]
BPD in Premature Infants	<20 ^1^	[100]
Postnatal Outcomes		
Respiratory Syncytial Infection in Children at First Year of Life	<20	[98]

^1^ 25(OH)D deficiency at 1 month of age. ^2^ In the mentioned studies for Table 1 and Table 2 the 25(OH)D concentration measurements were performed with various methods like ELISA, ECLIA, chemiluminescent assay, liquid chromatography coupled with mass spectrometry. Some of these methods, like ELISA, could give cross-reaction between 25(OH)D2 and 25(OH)D3 and could not distinguish these two forms (11). The difference between the kits and analysis equipment for chemiluminescent assays could also influence the precision of the obtained results. These data are presented in Appendix A.

## Data Availability

Not applicable.

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
