# Peer review of "Review: Influence of 25(OH)D Blood Concentration and Supplementation during Pregnancy on Preeclampsia Development and Neonatal Outcomes"

_ijms, 2022, doi:10.3390/ijms232112935_

Round 1
Reviewer 1 Report
Review:- influence of vitamin D blood concentration and supplementation during pregnancy on preeclampsia development and neonatal outcomes. [Karpova et al.]
This MS provides a review, as described in its title, that covers quite a lot of recent work and that concludes that supplementation at 1000 IU/day or increases in serum 25(OH)D of at least 10 ng/ml, [using, it is suggested, 50,000 IU doses of cholecalciferol once every 2 weeks] reduces the risk of preeclampsia and of other pregnancy complications and also reduces the risks of several adverse neonatal outcomes.
General comments.
1. There are several reviews that have been published in this area with regard to diet, calcium and vitamin D intakes already in 2022 and many more have appeared over recent years so that some comment on the differences in the conclusions reached would be useful. The fact that the present authors feel that they can justify making specific recommendations on improving maternal vitamin D status warrants added emphasis.
2. It is regrettable that that the text uses the term ‘vitamin D’ for the assays of circulating serum 25-hydroxyvitamin D [25(OH)D] that are currently used to assess vitamin D status, even in the title though vitamin D has not been measured in any of the studies included in this report. It would help readers, especially those new to this area, to use the correct term in the title and to make it clear early in the report just which abbreviation you are going to use to describe the 25(OH)D assay data throughout the study. This might be 25(OH)D, vitamin D status or some such abbreviation.
3. It would also be useful to know whether assays used in the many studies quoted were all specific for 25(OH)D3 and, hopefully, had no large cross-reactions with 25(OH)D2, and one also hopes that all the assay data used in the reports quoted had been measured in labs that were regularly in compliance with one of the widely used International Quality Control schemes, such as DEQUAS, so as to reassure readers that 25(OH)D data with unacceptable degrees of variability was not present in the studies used in the preparation of this report.
4. Readers might appreciate the provision of the equivalence of ng/ml to nmol/l early in the text.
Specific comments on the text [by Section, Paragraph and line number].
Section1, para 2 , line 3, why not say ‘25-hydroxyvitamin D concentrations [your abbreviation of choice]’ instead of ‘vitamin D concentrations’ here to deal with general comment 2.; para3, line 5, vitamin B must be a typo for vitamin D; para 6, line 1, ….. ‘ a threshold level for serum 25(OH)D concentration below which ….’ would be better than ‘after which’ since than phrase can be read to mean that higher values increase preeclampsia risks.
Section2, para 2, if the percentage of pregnant women with deficiency is high then all pregnant women should be supplemented routinely, and I suspect the evidence on this point would support routine supplementation as has been suggested to be necessary previously in many published references over the years; para 3 is part of the instructions to authors and should be removed;
Section 4. para 1, line 1, please define ‘bioavailable’ since both bound and free are bioavailable to various tissues and bound is specifically taken up through decidual/placental tissues as it is by renal tissues; Table 1, might it be useful to provide mean +/- SDs for the different trimesters, provided of course that you have checked that assay variance is not outside international QC standards in any of the studies quoted.[ see general comment 3].
Section 5, I take it the heading means ‘Effect of raising ……… so please say this
Section 6, para 1, line 1, proteinuria is a sign rather than a symptom I suspect, though women may report frothy urine. Line 12, an optimal would read better than the optimal since 25(OH)D values will vary between labs and assay methodologies and optimal values may well vary with population groups.
Section 7. Heading, this is more about associations than effects I suspect. Para 3, in view of the specific effects of diabetes on serum 25(OH)D and on calcitriol synthesis you should strengthen these remarks [see references in Bouillon & Bikle on the fall of Dogmas about vitamin D].
Section 9, last para, last line, the phrase ‘appointment’ of vitamin D supplements’ is not generally used where it is ‘provision’ that is being discussed.
Section 10, this MS does not mention or compare the usage of D3 and D2 which may not have been looked at much in pregnancy but in general these compounds induce effects somewhat differently and at different rates and it is generally accepted that it is D3 that should be used. This section would be a suitable place to refer to that aspect of vitamin D usage in brief, which could usefully be mentioned in passing in section 12.
Section 11. para 1, line 3, in the UK the 5 welfare foods given during WW2 included cod liver oil and eggs and rickets was virtually abolished; if there are reports on other outcomes in mothers and babies at that time they could be useful here; para 2 – re ref 137, did that report of bolus dosing show equally good benefits as daily supplementation. If so, this should be mentioned because the adverse effects of large bolus doses refer to boluses of ~100,000 IU and it is important for the maximum bolus doses that are fully effective to be identified, not just for rickets prevention but for other problems too, 100,000 IU doses having been reported to reduce hepatic 25-hydroxylation of vitamin D and to de-activate the enzyme producing activated vitamin D [calcitriol] through increased secretion of FGF23 and 100,000 IU doses having been found to be unable to prevent rickets; para 3, lines 1 onwards:- the D2d trial showed those benefits were only seen in prediabetes, with up to 70% reduction in risks of T2DM in those whose serum 25(OH)D values reached at least 100 nmol/l, [40 ng/ml] which only happened in those taking 4000 IU/day daily doses [and not on 3200 IU/day] [Dawson-Hughes et al, 2019] and you could usefully mention this here. I note that you report benefits from 4000 IU/day for neonates started pre pregnancy that were not seen in those on 4000 IU/day starting between weeks 10-18. These findings support the idea of supplementing all pregnant women routinely, preferably as part of pregnancy planning and without the delays caused by waiting for assay results, and these points could usefully be made both here and in the conclusions.
Section 12, para 2, you suggest supplementing when deficiency is detected but, as suggested above, wherever deficiency is common in pregnancy, [which is probably globally] supplementation for all pregnant women from early on in pregnancy, or as part of pregnancy planning as with folic acid, would be both more cost-effective and also more effective as it takes time to correct deficiency on the range of daily doses advised for pregnant women that you report. Taken together with the comments made on Section 11, I would hope to see these points mentioned in the Conclusions.
Author Response
- There are several reviews that have been published in this area with regard to diet, calcium and vitamin D intakes already in 2022 and many more have appeared over recent years so that some comment on the differences in the conclusions reached would be useful. The fact that the present authors feel that they can justify making specific recommendations on improving maternal vitamin D status warrants added emphasis.
- Indeed, in 2022 and earlier, a large number of literature reviews were published on vitamin D in the study of various pathologies. However, in our review, we consider both the main and applicable cases of D in the area of ​​preeclampsia. Therefore, the inclusion in the review of articles considering diet, calcium and vitamin D intakes outside the context of advertising, we consider it not relevant.
- It is regrettable that that the text uses the term ‘vitamin D’ for the assays of circulating serum 25-hydroxyvitamin D [25(OH)D] that are currently used to assess vitamin D status, even in the title though vitamin D has not been measured in any of the studies included in this report. It would help readers, especially those new to this area, to use the correct term in the title and to make it clear early in the report just which abbreviation you are going to use to describe the 25(OH)D assay data throughout the study. This might be 25(OH)D, vitamin D status or some such abbreviation.
- Thanks for the helpful note. We have corrected the abbreviation for circulating serum 25-hydroxyvitamin D [25(OH)D] in the article title and all text.
- It would also be useful to know whether assays used in the many studies quoted were all specific for 25(OH)D3 and, hopefully, had no large cross-reactions with 25(OH)D2, and one also hopes that all the assay data used in the reports quoted had been measured in labs that were regularly in compliance with one of the widely used International Quality Control schemes, such as DEQUAS, so as to reassure readers that 25(OH)D data with unacceptable degrees of variability was not present in the studies used in the preparation of this report.
- We have added information about the method of measuring the concentration of 25(OH)D3 in supplementation.
Considering measurements of 25(OH)D3, studies mentioned in table 1 and table 2 majorly analyzed measurements of 25(OH)D, which included 25(OH)D3 and 25(OH)D2 (except reference 79). Only a few studies followed the International Quality Control DEQUAS (47, 48, 49, 50, 79), which gives the risk of great variability of 25(OH)D data presented in our review. We now addressed this issue in the text and stated to the readers about the uncertainty of comparison of obtained data. Furthermore, it was also added that 25(OH)D concentration analysis was performed with different methods (mostly ELISA and Liquid Chromatography coupled with Mass Spectrometry), which could give cross-reaction between 25(OH)D2 and 25(OH)D3 or even not enabled distinguish these two forms. In some articles the interassay coefficient of variation for the ELISA was 10.3%. The ELISA recognized 100% of 25(OH)D3 and 75% of 25(OH)D2 but did not distinguish between these two forms. In our initial HPLC validation, we observed that only three of 32 samples (<10%) had any measurable 25(OH)D2, and within these samples, 25(OH)D2 accounted for only 10% of the total measurable 25(OH)D.
4 Readers might appreciate the provision of the equivalence of ng/ml to nmol/l early in the text.
- We added the equivalence of ng/ml to nmol/l early in the second section at first mention 25(OH)D concentration.
Reply to Specific comments on the text
Section1, para 2 , line 3, why not say ‘25-hydroxyvitamin D concentrations [your abbreviation of choice]’ instead of ‘vitamin D concentrations’ here to deal with general comment 2.; para3, line 5, vitamin B must be a typo for vitamin D; para 6, line 1, ….. ‘ a threshold level for serum 25(OH)D concentration below which ….’ would be better than ‘after which’ since than phrase can be read to mean that higher values increase preeclampsia risks.
- Changes made
Section2, para 2, if the percentage of pregnant women with deficiency is high then all pregnant women should be supplemented routinely, and I suspect the evidence on this point would support routine supplementation as has been suggested to be necessary previously in many published references over the years;
- Not all publications note whether pregnant women have taken 25(OH)D3 supplementsÑŽ. For this reason, we review the role of 25(OH)D3 supplementation in reducing the risk of preeclampsia separately in Sections 11 and 12
para 3 is part of the instructions to authors and should be removed;
- Changes made
Section 4. para 1, line 1, please define ‘bioavailable’ since both bound and free are bioavailable to various tissues and bound is specifically taken up through decidual/placental tissues as it is by renal tissues; Table 1, might it be useful to provide mean +/- SDs for the different trimesters, provided of course that you have checked that assay variance is not outside international QC standards in any of the studies quoted.[ see general comment 3].
- Since 25(OH)D concentration measurements were carried out by different methods in the articles under review. Therefore, we cannot compare or compare only data from chromatography.
Section 5, I take it the heading means ‘Effect of raising ……… so please say this
- In Section 5, we consider the effect of maternal blood 25(OH)D3 concentrations on blood pressure. The use of supplements containing 25(OH)D3 has been cited as evidence of a correlation between blood pressure and 25(OH)D3 concentrations. Now we have corrected this section and changed its title:
- Impact of 25(OH)D3 Concentrations on the development of arterial hypertension during pregnancy complicated by preeclampsia
25(OH)D3 plays an essential role in regulation of blood pressure (BP), affecting vascular endothelial function, oxidative stress and placenta [143, 34]. In addition, vitamin D regulates key target genes associated with placental invasion, normal implantation, and angiogenesis [42, 34]. Thus, the level of 25(OH)D3 may be one of the causes of arterial hypertension in preeclampsia. According to several studies, low concentrations of 25(OH)D3 and 1,25(OH)2D correlate with high systolic and diastolic BP during pregnancy [58-63] and only a few papers have demonstrated opposite results [64]. Because 25(OH)D3 is an approved supplement during pregnancy, Nassar, Seham Zakaria, and Noha Mohamed Badae tested the hypothesis of lowering systolic BP by 25(OH)D3 supplementation and found that its administration reduced systolic BP, proteinuria [65]. Thus, there is strong evidence for the need to measure vitamin D concentrations among pregnant women with any form of hypertension to prescribe optimal doses of 25(OH)D3, thereby possibly providing a reduction in elevated systolic and diastolic blood pressure, reducing the risk of additional complications for the mother and fetus.
Section 6, para 1, line 1, proteinuria is a sign rather than a symptom I suspect, though women may report frothy urine.
- Proteinuria is observed as a symptom of preeclampsia, according to the clinical recommendation :Hypertensive disorders during pregnancy, childbirth and the postpartum period. Preeclampsia. Eclampsia. Russian clinical guidelines (In Russ.), 2016, p.5
Section 7. Heading, this is more about associations than effects I suspect. Para 3, in view of the specific effects of diabetes on serum 25(OH)D and on calcitriol synthesis you should strengthen these remarks [see references in Bouillon & Bikle on the fall of Dogmas about vitamin D].
- We agree that the word “effect '' does not fully characterize this chapter. Because of it we changed chapter name on: “Relationship between Impact 25(OH)D Concentrations and the Risk of Preeclampsia in Diabetes Mellitus” The Bouillon & Bikle study on the fall of Dogmas about vitamin D looks at animal modeling results, which we do not include in the review because we only look at human blood 25(OH)D concentrations. We will consider animal models in a separate review, which is at the stage of writing.
Section 9, last para, last line, the phrase ‘appointment of vitamin D supplements’ is not generally used where it is ‘provision’ that is being discussed.
- - Changes made
Section 10, this MS does not mention or compare the usage of D3 and D2 which may not have been looked at much in pregnancy but in general these compounds induce effects somewhat differently and at different rates and it is generally accepted that it is D3 that should be used. This section would be a suitable place to refer to that aspect of vitamin D usage in brief, which could usefully be mentioned in passing in section 12.
- All of the studies we reviewed were prescribing vitamin D3 supplements in the context of preeclampsia, so consideration of vitamin D2 supplementation is not applicable to our article.
Section 11. para 1, line 3, in the UK the 5 welfare foods given during WW2 included cod liver oil and eggs and rickets was virtually abolished; if there are reports on other outcomes in mothers and babies at that time they could be useful here; para 2 – re ref 137, did that report of bolus dosing show equally good benefits as daily supplementation. If so, this should be mentioned because the adverse effects of large bolus doses refer to boluses of ~100,000 IU and it is important for the maximum bolus doses that are fully effective to be identified, not just for rickets prevention but for other problems too, 100,000 IU doses having been reported to reduce hepatic 25-hydroxylation of vitamin D and to de-activate the enzyme producing activated vitamin D [calcitriol] through increased secretion of FGF23 and 100,000 IU doses having been found to be unable to prevent rickets; para 3, lines 1 onwards:- the D2d trial showed those benefits were only seen in prediabetes, with up to 70% reduction in risks of T2DM in those whose serum 25(OH)D values reached at least 100 nmol/l, [40 ng/ml] which only happened in those taking 4000 IU/day daily doses [and not on 3200 IU/day] [Dawson-Hughes et al, 2019] and you could usefully mention this here. I note that you report benefits from 4000 IU/day for neonates started pre pregnancy that were not seen in those on 4000 IU/day starting between weeks 10-18. These findings support the idea of supplementing all pregnant women routinely, preferably as part of pregnancy planning and without the delays caused by waiting for assay results, and these points could usefully be made both here and in the conclusions.
- It would be very educational and resourceful to include other neonatal outcomes connected with 25(OH)D3 insufficiency in a separate review focused solely on that topic. Our paper is concentrated on the connection of 25(OH)D3 deficiency and preeclampsia and associated with its outcomes, which were mentioned and discussed in this section.
- We have added information why we believe it is important to prescribe vitamin D supplements only after a 25(OH)D concentration test before planning or during pregnancy: It's important to consider that routine doses of vitamin D may be insufficient to achieve optimal levels (>30 ng/ml), and excessively high vitamin D concentrations may be hazardous to both maternal and fetal health. For this reason, it is suggested to measure vitamin D levels prior to prescribing a supplement to determine the appropriate dosage for each individual case.
Line 12, an optimal would read better than the optimal since 25(OH)D values will vary between labs and assay methodologies and optimal values may well vary with population groups.
- Since the concept of the optimal concentration of 25(Ap)B3 differs depending on the method of analysis: laboratory and population: we have clarified what concentrations of vitamin D we consider to be optimal: In addition, the optimal 25(OH)D3 (> 30 ng/ml) concentrations before pregnancy can have a small impact on immunity and embryogenesis, while with an initial deficiency, additional loss of 25(OH)D3 can cause serious consequences, especially for fetal development.
Reviewer 2 Report
This is a very interesting and generally well written review but it presents some flaws that must be resolved. In particular:
Introduction: Authors properly illustrated the clinical manifestation of preeclampsia but, since this a review, it deserves to be also pointed out the placenta dysfunctions associated with this pathology. In fact, PE pregnancies are also characterised by trophoblast immaturity (PMID: 32529396) and vascular dysfunction (PMID: 34831277). This is an important point to add since these are two very important features of this disease and Vitamin D deficiency could play an important role in these placental dysfunction.
2. Vitamin D concentrations in the first trimester of pregnancy (<14) complicated by preeclampsia: correct with (<14 weeks)
5. Effect of Vitamin D Concentrations on Hypertensive Disorders in Pregnancy: Authors must give a definition of Hypertensive Disorders in Pregnancy. In particular, they must describe what types of disorders are included.
Acronyms must be written in full length when mentioned for the first time
Author Response
Good afternoon! We are grateful to you for your valuable comments. It helped us to get a clearer understanding of how to make our work more educational and meet the standards of the academic community.
- We have added more information about hypoxia, endothelial and vascular dysfunction. The role of 25(OH)D3 in the development of endothelial dysfunction is more emphasized in our text now. However, this topic is still not fully discussed in our paper as a separate review is being prepared on this issue in the context of preeclampsia.
The etiology and pathogenesis of preeclampsia are not fully understood. Placental hypoxia and/or ischemia, high levels of oxidative stress, endothelial dysfunction and trophoblast immaturity are the hallmarks of preeclampsia. Moreover, vascular reactivity is caused by an imbalance and malfunction in the chemicals that cause vasodilation and vasoconstriction. The ensuing vasoconstriction reduces mother organ perfusion and placental perfusion, which might harm end organs. There are potential epigenetic, biochemical and biophysical biomarkers for predicting the development of this pathology. Numerous studies have confirmed the contribution of 25-hydroxyvitamin D (25-Hydroxycalciferol, 25(OH)D3) concentrations to development of preeclampsia and associated complications, such as endothelial and vascular dysfunction [2-3, 9–25].
- Section title corrected: 25(OH)D3 concentrations in the first trimester of pregnancy (<14 weeks) complicated by preeclampsia.
- Initially we considered hypertension as a symptom of preeclampsia. In order to clarify our point, we changed the title of the section and elaborated our statement. Impact of 25(OH)D3 Concentrations on the development of arterial hypertension during pregnancy complicated by preeclampsia.
25(OH)D3 plays an essential role in regulation of blood pressure (BP), affecting vascular endothelial function, oxidative stress and placenta [143, 34]. In addition, vitamin D regulates key target genes associated with placental invasion, normal implantation, and angiogenesis [42, 34]. Thus, the level of 25(OH)D3 may be one of the causes of arterial hypertension in preeclampsia. According to several studies, low concentrations of 25(OH)D3 and 1,25(OH)2D correlate with high systolic and diastolic BP during pregnancy [58-63] and only a few papers have demonstrated opposite results [64]. Because 25(OH)D3 is an approved supplement during pregnancy, Nassar, Seham Zakaria, and Noha Mohamed Badae tested the hypothesis of lowering systolic BP by vitamin D supplementation and found that its administration reduced systolic BP, proteinuria [65]. Thus, there is strong evidence for the need to measure 25(OH)D3 concentrations among pregnant women with any form of hypertension to prescribe optimal doses of 25(OH)D3, thereby possibly providing a reduction in elevated systolic and diastolic blood pressure, reducing the risk of additional complications for the mother and fetus.
- We wrote Acronyms in full at the first mention